# Perfusion Parameters in Near-Infrared Fluorescence Imaging with Indocyanine Green: A Systematic Review of the Literature

**DOI:** 10.3390/life11050433

**Published:** 2021-05-11

**Authors:** Lauren N. Goncalves, Pim van den Hoven, Jan van Schaik, Laura Leeuwenburgh, Cas H. F. Hendricks, Pieter S. Verduijn, Koen E. A. van der Bogt, Carla S. P. van Rijswijk, Abbey Schepers, Alexander L. Vahrmeijer, Jaap F. Hamming, Joost R. van der Vorst

**Affiliations:** Department of Surgery, Leiden University Medical Center, Albinusdreef 2, 2300 RC Leiden, The Netherlands; laurengoncalves9@gmail.com (L.N.G.); P.van_den_Hoven@lumc.nl (P.v.d.H.); J.van_Schaik@lumc.nl (J.v.S.); L.C.Leeuwenburgh@lumc.nl (L.L.); cas.hendricks@hotmail.com (C.H.F.H.); P.S.Verduijn@lumc.nl (P.S.V.); K.E.A.van_der_Bogt@lumc.nl (K.E.A.v.d.B.); C.S.P.van_Rijswijk@lumc.nl (C.S.P.v.R.); A.Schepers@lumc.nl (A.S.); a.l.vahrmeijer@lumc.nl (A.L.V.); J.F.Hamming@lumc.nl (J.F.H.)

**Keywords:** perfusion imaging, near infrared fluorescence, indocyanine green

## Abstract

(1) Background: Near-infrared fluorescence imaging is a technique capable of assessing tissue perfusion and has been adopted in various fields including plastic surgery, vascular surgery, coronary arterial disease, and gastrointestinal surgery. While the usefulness of this technique has been broadly explored, there is a large variety in the calculation of perfusion parameters. In this systematic review, we aim to provide a detailed overview of current perfusion parameters, and determine the perfusion parameters with the most potential for application in near-infrared fluorescence imaging. (2) Methods: A comprehensive search of the literature was performed in Pubmed, Embase, Medline, and Cochrane Review. We included all clinical studies referencing near-infrared perfusion parameters. (3) Results: A total of 1511 articles were found, of which, 113 were suitable for review, with a final selection of 59 articles. Near-infrared fluorescence imaging parameters are heterogeneous in their correlation to perfusion. Time-related parameters appear superior to absolute intensity parameters in a clinical setting. (4) Conclusions: This literature review demonstrates the variety of parameters selected for the quantification of perfusion in near-infrared fluorescence imaging.

## 1. Introduction

Near-infrared fluorescence (NIRF) imaging is a promising technique for visualizing tissue perfusion. The measurement of fluorescence in the near-infrared spectrum is feasible for perfusion assessment due to the low tissue auto-fluorescence in this range. This allows for the visualization of an intravenously administered fluorophore. The most frequently used fluorescent dye in perfusion assessment is indocyanine green (ICG), which is primarily contained in the vascular system due to its selective binding with the plasma protein albumin. NIRF imaging is minimally invasive, with a low risk of side-effects. The use of NIRF imaging as a technique to assess tissue perfusion has been explored in various surgical fields. In plastic surgery, this technique has been used to assess perfusion in flap surgery, nipple-sparing mastectomies, and reconstructive microsurgery [1]. NIRF imaging has been applied in endocrinological surgery, assisting in the preservation of critical structures, such as the parathyroid gland, during surgical procedures. For patients with peripheral arterial disease (PAD), NIRF imaging can be used for the assessment of regional tissue perfusion. Described applications of NIRF imaging in patients with PAD include (1) diagnosis, (2) the measuring of the effect of revascularization procedures, and (3) the assessment of tissue viability following amputation surgery [2]. In cardiac interventions, the use of NIRF imaging has allowed for the real-time assessment of graft patency, with applications in transplantation surgery providing a diagnostic tool for the assessment of kidney microperfusion, for example [3]. NIRF imaging has, furthermore, been applied in neurosurgery, quantifying cerebral perfusion, both intra- and postoperatively, and in gastrointestinal surgery, assessing anastomotic perfusion after bowel resection [4].

To date, the perfusion patterns visualized with fluorescence imaging have been quantified using time–intensity curves, from which various parameters can be extracted for statistical analysis. This article will systematically review the literature on the time–intensity curve parameters following NIRF imaging in perfusion assessment, with the aim of identifying optimal perfusion parameters for the standardized quantification of perfusion using NIRF imaging.

## 2. Materials and Methods

### 2.1. Search Strategy

An electronic search was conducted using Pubmed, Medline, Embase, and Cochrane Review from inception to January 2021 to identify all relevant literature. Gray literature was not included. Medical subject headings (MeSH) were adopted, and included: “perfusion”, “near-infrared fluorescence imaging”, and “indocyanine green”. The search strategies applied can be found in the Appendix B. A manual search of the references of the included articles was also performed to identify further studies of interest. Articles were systematically screened according to the PRISM guidelines with a two-stage method, with stage one including the screening of the title and abstract, followed by stage two, with full-text screening. The systematic review of the literature and results conducted in this article are not part of a registered study.

### 2.2. Article Selection

Only full-text articles in English were included. Articles reporting the use of fluorescence imaging in animal studies were excluded. Article selection was performed by two independent researchers (L.G. and P.H.). Any article discussing the analysis of tissue perfusion using near-infrared fluorescence imaging was included if fluorescence–intensity curves and perfusion parameters were mentioned. Articles reporting the qualitative use of NIRF imaging without quantitative data were excluded. NIRF perfusion imaging has been applied in numerous surgical fields. The results obtained are therefore divided by subspecialization, which included gastrointestinal, neurological, vascular, transplantation, and plastic surgery, as well as other surgical subspecializations.

### 2.3. Quality Assessment

The quality and the risk of bias of the selected and included articles were independently evaluated by two reviewers (L.G. and P.H.) according to the revised quality assessment of diagnostic accuracy studies [5]. In instances of discrepancy in interpretation, a third independent reviewer (JV) was asked to adjudicate.

### 2.4. Data Extraction

Data extracted from all of the articles that were reviewed for inclusion consisted of the relevant patient characteristics, the type of NIRF imaging camera, the ICG concentration, and the perfusion parameters selected.

## 3. Results

An overview of the article selection process for this systematic review is reported in a flow diagram in Figure 1, according to the preferred reporting items for systematic review and meta-analysis protocols’ 2015 guidelines. A total of 1511 articles were found based on the search terms in Appendix B, of which 113 were suitable for review, with a final selection of 57 articles. Manual review of the references in the articles selected above led to the inclusion of 2 further references, providing a total of 59 articles for final inclusion. The results of the quality assessment can be found in the Appendix A. The 59 studies selected included a total of 2336 patients. The number of patients in the studies ranged from 1 to 181. All studies in this review used ICG as a fluorescence marker. The selected studies were divided into the following fields: vascular surgery (*n* = 18), gastrointestinal surgery (*n* = 8), plastic surgery (*n* = 15), neurosurgery (*n* = 14), and transplantation surgery (*n* = 2). Furthermore, one diabetes mellitus study, one thyroid surgery study, and one study regarding the imaging of breast lesions were also included. All studies were published during the period from 2008 to 2021. An overview of the perfusion parameters mentioned in the literature is reported in Table 1.

A large selection of NIRF imaging systems were used, the most common were the SPY Elite system (*n* = 12), Flow 800 (*n* = 13), and the photodynamic eye imaging system (*n* = 13).

The reviewed articles mentioned a total of 26 perfusion parameters distilled from fluorescence-intensity curves. All of the reviewed articles selected a unique combination of the perfusion parameters in Table 1 and schematically shown in Figure 2 and Figure 3.

NIRF imaging was performed either pre-, post-, or intraoperatively. Time-dependent parameters were described in all the included literature. Perfusion parameters were based on:(1)Absolute fluorescence intensity;(2)Time;(3)Changes in intensity over time.

### 3.1. Absolute Intensity Parameters

Multiple studies described significant changes in intensity-related parameters. The maximum fluorescence intensity (Imax) was the most frequently mentioned parameter (*n* = 29). Fluorescence intensity parameters also included ingress, egress, fluorescence intensity as a range in arbitrary units, and the fluorescence intensity at the end of the measurement period or study, as described in Table 1.

### 3.2. Inflow and Outflow Parameters

Inflow parameters could be calculated from the upslope segment of the time–intensity curve, and could be correlated with tissue perfusion. These included time-specific parameters such as the Tmax, T1/2, time to peak intensity, and rise time (Figure 2 and Figure 3). Once ICG had been intravenously administered, an initial fluorescence signal could be detected (Tstart). Following the detection of fluorescence intensity, the time taken to reach 50% of the maximum intensity (T1/2max) and the peak intensity (Tmax) could be calculated. Furthermore, during the upslope, the interaction between the fluorescence intensity and time was also broadly explored, and included the ingress rate, or the blood flow index, which were all ratios or rates of increase in the fluorescence signal over time.

Outflow parameters were calculated to quantify the decrease in ICG intensity over time, thus providing information on vascular elimination. Time-specific outflow parameters included the intrinsic transit time (ITT), which is the time needed for ICG to circulate from the arterial to the venous system. The downslope interaction between intensity and time was quantified by the egress rate and IR 60, which is the intensity at 60 s after Tmax/Fmax, for example.

### 3.3. Relative Parameters

Relative parameters were also calculated by selecting and comparing two or more regions of interest, for example, a region with suspected ischemia being compared with another region with optimal perfusion, the reference region. The relative parameters were calculated by dividing the targeted perfusion value by the reference value.

### 3.4. Gastro-Intestinal Surgery

The surgical procedures discussed in this review include colorectal surgery (*n* = 6), specifically resections followed by anastomosis, and esophagectomy (*n* = 2). Two of the six studies on colorectal surgery performed a retrospective analysis of the prospective data [32,54]. Four of the eight studies on gastrointestinal surgery included in this review administered a weight-dependent dose of ICG, as noted in Table 2 [25,26,54,56].

The Fmax was researched in six studies (*n* = 335) [25,26,28,31,32,47]. Four studies described no significant differences in the Fmax values between the groups with and without anastomotic leakage in both colorectal and esophageal surgery [25,26,28,32]. Wada et al. (*n* = 112) highlighted the predictive value of Fmax for anastomotic leakage in receiver operator curve (ROC) analysis, with a sensitivity and specificity of 100% and 92%, respectively, at a cut-off of 52.0 units [47]. The Fmax did not significantly predict early or late flatus or defecation. A study by Ishige et al. examined the application of ICG-NIRF imaging in quantifying the perfusion of the gastric conduit in esophagectomy [31]. A relative comparison of Fmax in a control phase, gastric tube phase, and anastomotic phase prior to intrathoracic or cervical esophago-gastronomy was calculated and shown to be significantly different between phases, although no anastomotic leakage occurred.

Fmin was a parameter selected by Son et al. (*n* = 86), and was described as the fluorescence intensity at the baseline [25]. There was no significant difference between the groups with and without anastomotic leakage.

Five studies included the Tmax as a perfusion parameter (*n* = 245) [26,28,31,47,56] D’Urso et al. highlighted a statistically significantly lengthened Tmax in both the proximal and distal colorectal resection sites in the complications group in comparison to the uncomplicated cases (*p* = 0.01) [56]. Furthermore, correlation to clinical parameters such as intestinal lactate and mitochondrial efficiency showed varying significance depending on the region and parameter selected [56]. Amagai et al. described a significant correlation between the Tmax and anastomotic leakage in one of the four selected regions of interest (*p* = 0.015), while Hayami et al. mentioned no significant correlation to anastomotic leakage [26,28]. The predictive value of the Tmax in clinical outcomes such as early or late flatus or defecation was significant in one study (*n* = 112) (*p* = 0.02, *p* = 0.01, respectively) [47]. Son et al. and Amagai et al. described the Tmax as the difference between the initial fluorescence intensity and the maximum fluorescence intensity [25,26]. Both studies highlighted a statistically significant difference in the Tmax values between the anastomotic leakage groups (Son *p* < 0.001, Amagai *p* = 0.015).

The inflow parameter T0, defined as the time to the initial fluorescence signal, was studied by Aiba et al. and Hayami et al. [28,54]. Both of the aforementioned studies showed T0 to significantly differ with regard to anastomotic leakage (Aiba *p* = 0.046, Hayami, *p* = 0.0022).

The TR, time ratio, is a parameter encompassing the ratio between T1/2 and Tmax, and was shown to significantly differ between the anastomotic leakage and no anastomotic leakage groups (*n* = 86) [25]. Furthermore, a ROC analysis with an area under the curve higher than 0.9 suggested that a TR of 0.6 is significantly predictive of anastomotic leakage [25].

T1/2 was adopted as an inflow parameter by four studies (*n* = 246) [25,28,32,47]. Two studies by Kamiya et al. and Son et al. described a statistically significant difference in T1/2 in the anastomotic failure or leakage groups (*p* < 0.01, *p* < 0.001, respectively) [25,32]. One study showed no significant difference between the groups with and without anastomotic leakage [28].

The slope was investigated in two studies (*n* = 134) [28,47]. No significant intraoperative difference or prediction of postoperative outcomes was described.

### 3.5. Neurosurgery

Fourteen studies (*n* = 345) investigated the application of ICG angiography in vascular neurosurgery [27,33,34,35,41,45,48,49,58,59,61,62,63,64]. Eight of the fourteen studies selected for dose-dependent ICG administration, with doses ranging from 0.1 mg/kg to 0.3 mg/kg, see Table 2 for further details. The studies analyzed the following inflow parameters: Tmax, T1/2, RT, slope, transit time, and cerebral blood flow index. The rise time is heterogeneously defined as the interval between 10% and 90% of the signal, or 20% and 80% of the signal.

The Tmax parameter was examined in seven studies (*n* = 191) [27,33,34,35,41,49,64]. One study found the Tmax to significantly discriminate between patients with impaired and normal cerebral perfusion in the occlusive cerebral arterial disease and control groups (*p* = 0.013) [35]. Furthermore, the ratio of Tmax pre- and post-bypass procedure was significantly lower in patients with postoperative hyperperfusion syndrome than in patients without postoperative hyperperfusion syndrome (*p* = 0.017) [35].

The T1/2 was explored in four articles (*n* = 181) [27,48,58,59]. Three studies researched the application of this perfusion parameter in patients pre- and post-bypass surgery, with only one study by Prinz et al. reporting a significant decrease (*p* = 0.001) [27,58,59]. However, when Prinz et al. compared the ICG perfusion data to the quantitative Doppler flow, there were no significant correlations between the diagnostic methods.

Four articles (*n* = 44) selected rise time (RT) as an ICG perfusion parameter [33,35,45,61]. Holling et al. found RT to be significantly shorter following a bypass procedure (*p* = 0.025), with no change between the pre- and post-bypass data in the control measurements (*p* = 0.125) [61]. A study by Kamp et al. exploring cortical perfusion following traumatic brain injury found the arterial RT to be significantly longer in patients with a favorable outcome at 3 months (*p* = 0.002).

Of the four studies (*n* = 153) that calculated transit time as a perfusion parameter, two studies by Holling et al. and Ye et al. described significant results [27,41,48,61]. Ye et al. highlighted a significant difference in arteriovenous transit time in a heterogenous group of patients including arteriovenous malformations, moyamoya disease, and both unruptured and ruptured cerebral aneurysms [48].

Cerebral blood flow (CBF), or slope, was the most commonly selected neurovascular perfusion parameter, mentioned in 10 articles (*n* = 256) [34,35,41,45,48,49,58,59,63,64]. Seven studies examined the change in cerebral blood flow pre- and post-bypass procedure [35,41,49,58,59,63,64]. A statistically significant change in CBF was noted in five of the aforementioned seven studies [35,41,58,63,64]. Four studies also examined the relationship between the development of postoperative hyperperfusion and CBF, with two studies by Zhang et al. and Uchino et al. describing a significant increase in CBF in the symptomatic hyperperfusion group (*p* < 0.001, *p* = 0.013, respectively) [35,49,63,64].

One study by Kamp et al. described the parameter residual fluorescence intensity as a percentage of maximum fluorescence intensity in patients with traumatic brain injury [33]. The cortical and venous residual fluorescence intensity was significantly higher in the cortical and venous tissue (*p* = 0.01, *p* = 0.02, respectively) in patients with an unfavorable outcome. No significant difference was noted in arterial residual fluorescence intensity (*p* = 0.05).

Absolute fluorescence intensities were calculated in nine studies [27,33,34,35,41,45,48,49,64]. Five studies found a significantly different Imax value between the perfusion groups [35,45,48,49,64]. Kobayashi et al. noted a significant increase in Imax after STA-MCA bypass surgery (*p* = 0.047), yet no significant differences in the Imax values when comparing patients with and without postoperative hyperperfusion [35]. However, Zhang et al. also investigated Imax as a perfusion parameter for the detection of postoperative hyperperfusion, and described a significantly higher Imax in the symptomatic hyperperfusion group (*p* < 0.001) [49].

### 3.6. Plastic Surgery

Fluorescent imaging was quantitatively reported in 15 articles, including 515 patients [6,7,10,13,17,19,22,23,36,37,46,51,52,53,57]. The fields of application included free-flap and DIEP flap perfusion, breast reconstruction surgery, and microvascular surgery. The SPY Elite Imaging system was the most commonly selected (*n* = 8), see Table 2.

*Flap perfusion.* Flap perfusion studies examined free-flap surgery, flap perfusion in maxillofacial surgery, and DIEP flap procedures. Relative perfusion was researched in three studies [6,7,52]. Abdelwahab et al. showed the flap-to-cheek ratio to be statistically significant in predicting flap vascularization, while Betz et al. highlighted no significant differences in the relative slope values in free-flap surgery [6,7,52].

Inflow and outflow parameters were mentioned in seven (*n* = 128) free-flap and DIEP flap studies [10,13,23,36,37,52,53]. Two studies describe no significant differences in the ingress and ingress rate [10,13]. Significant differences in inflow parameters including the slope, Tmax, PDE10, slope at T1/2, and T1/2 were seen in three studies [23,37,53]. Hitier et al. described a significantly lower per-operative Fmax in flaps with vascular complications (*p* = 0.008) [23]. Furthermore, in flaps with a significantly lower postoperative slope (*p* = 0.02) and Fmax (*p* = 0.03), the values returned to normal following surgical revision [23]. One study examining single-pedicle versus bipedicle blood supply in flap reconstruction found no significant difference in the Imax between the aforementioned groups, with a significant difference in the Tmax, T1/2, slope, PDE10, and slope at T1/2 between the same groups (*p* < 0.05) [37].

Absolute intensity parameters were not significant in the studies by Betz et al. and Miyazaki et al. [37,53].

### 3.7. Breast Reconstructive Surgery

Gorai et al. explored the use of relative ICG perfusion parameters in the prediction of skin necrosis following tissue expander reconstruction in breast cancer patients [57]. The application of ICG imaging led to a significantly lower rate of necrosis (*p* < 0.05), with a statistically significant difference in relative perfusion (*p* < 0.001). Yang et al. evaluated mastectomy flap perfusion at various tissue expander volumes, with significant differences in the ingress and egress between the different expander volume groups (*p* = 0.0001, *p* = 0.0037, respectively) [17].

### 3.8. Microvascular Surgery

Reconstructive surgical procedures are dependent on adequate vascular and nervous supply to the operated region. Tanaka et al. explored the use of ICG imaging to detect blood supply to the femoral cutaneous and vastus lateralis motor nerve, allowing for the selection of the better vascularized nerve [46]. The selected inflow parameters, slope and Tstart, were heterogeneously statistically significant, while the Tmax showed no statistical significance in any region measured [46]. Fichter et al. examined the effect of the number of osteotomies on bone perfusion in free fibula flaps [19]. The study found a significant difference in the slope with additional osteotomies (*p* = 0.034). Mothes et al. explored tissue perfusion during hand revascularization surgery, and found the intraoperative slope values to significantly differ in tissue that survived postoperatively (*p* < 0.01) [51]. Three studies (*n* = 75) researched absolute fluorescent intensity, namely the Imax, reporting heterogenous results depending on time points or regions selected for comparison [22,46,51].

### 3.9. Vascular Surgery

Quantitative analysis of fluorescence imaging was conducted in 18 studies included in this review (*n* = 683) [8,11,12,15,16,18,20,29,30,38,39,40,43,44,50,55,60,65]. Fifteen studies opted for dose-dependent ICG administration, with 10 studies selecting a dose of 0.1 mg/kg [12,15,16,20,29,30,38,39,40,43,44,50,55,60,65]. SPY Elite Imaging was the fluorescence imaging system of choice in eight studies, see Table 2. The fields of application within vascular surgery included patients with PAD, patients receiving dialysis, prediction of wound healing, microperfusion following arteriovenous fistula formation in dialysis patients, and perfusion in Raynaud’s phenomenon.

The Fmax was the most frequently selected parameter (7 studies; *n* = 297) [18,29,30,38,39,40,50]. Four studies described a significant difference in the Fmax pre- and post-revascularization, with Nakamura et al. mentioning no significant difference in the treated limb, and a significant decrease in the contralateral limb (*p* = 0.0875, *p* = 0.006, respectively) [18,29,38,40]. No significant difference in Fmax was detected between PAD patients and controls; dialysis patients and controls; Rutherford classification categories; or critical limb ischemia patients and controls [30,39,50].

Six studies selected the ingress and ingress rate as perfusion parameters (*n* = 237) [8,12,15,16,18,40]. Five studies evaluated the ingress in PAD patients, all describing statistically significant results pre- and post-revascularization [8,15,16,18,40]. Regus et al. highlighted a statistically significant difference in the ingress and ingress rate pre- and post-arteriovenous anastomotic creation in the hand and fingers of dialysis patients (*p* < 0.001) [12]. Furthermore, there was a significant difference in the intraoperative ingress ratio and ingress rate ratio in patients who had developed hemodialysis access-induced distal ischemia (*p* = 0.001, *p* = 0.003, respectively) [12].

The egress and egress rate provided heterogenous results in PAD patients undergoing revascularization procedures in two studies [8,18]. One study by Braun et al. showed a significant difference in both the egress and egress rate pre- and post-revascularization (*p* = 0.004, *p* = 0.013, respectively), while there was no significant difference in the egress in a study by Colvard et al. (*p* = 0.35) [8,18].

The perfusion parameter Tmax was discussed in six studies (*n* = 204) [29,30,38,39,40,55]. Heterogenous results were seen in PAD patients pre- and post-revascularization. Igari et al. described a significant difference in the Tmax in three regions of interest following revascularization [29]. A study by Nakamura et al. also found the Tmax to be significantly different post-revascularization, in both the intervention limb and the contralateral limb (*p* = 0.016, *p* = 0.013, respectively) [38]. Only one study examined the Tmax in PAD and control patients, with a significant difference between the groups (*p* < 0.05) [30].

Six studies selected the T1/2 as a perfusion parameter [29,30,38,39,60,65]. One study by Venermo et al. showed no significant different in T1/2 between patients with and without diabetes [65]. Igari et al. stated no significant differences between PAD patients and the controls [30]. Two studies described significant changes in the T1/2 in PAD patients pre- and post-revascularization [29,38].

Studies exploring the T1/2 in patients with diabetes who were on dialysis and grouped according to Fontaine classification or critical limb ischemia showed heterogenous results [39,60,65].

Two studies (*n* = 51) examined the slope [29,50]. One study showed a significant change in the slope in three selected regions of interest following revascularization in PAD patients (*n* = 21) [29]. A study by Zimmermann et al. described a significant difference in the slope in patients, grouped according to the Rutherford classification, with an increased extent of arterial collateralization and in patients with critical limb ischemia (*p* < 0.001, *p* = 0.005, *p* < 0.001) [50].

The PDE10, described as the fluorescence intensity increase at 10 s, was examined in four studies [29,44,60,65]. Two studies stated a statistically significant change in the PDE10 in PAD patients following a revascularization procedure [29,44]. An ROC analysis was performed in two studies, with a cut-off PDE10 of 28 s at a transcutaneous pressure (TcPO2) of 30 mmHg in a study by Terasaki et al., and a cut-off PDE10 of 21 arbitrary units at a TcPO2 of 40 mmHg in a study by Venermo et al. [60,65].

Eleven of the eighteen vascular surgery studies included in this review assessed the correlation between the aforementioned perfusion parameters and standard diagnostic methods in this field such as the ankle–brachial index (ABI), toe pressure, TcPO2, and the toe–brachial index (TBI) [8,11,18,29,30,38,40,43,44,50,65]. Four studies found no significant correlation between the ABI and ICG-NIRF perfusion parameters [11,38,40,44]. Significant but heterogenous correlation was seen between the ABI and a range of parameters in five studies [8,18,29,30,65].

### 3.10. Transplantation

Two studies (*n* = 205) explored the application of quantitative ICG imaging in transplantation surgery [9,14]. Both studies utilized the SPY Elite Imaging System and an ICG dose of 0.02 mg/kg. The inflow parameters of ingress, ingress rate, egress, and egress rate were assessed. Rother et al. and Gerken et al. described the ingress and ingress rate to significantly detect differences in kidney perfusion. Gerken et al. explored the association of intraoperative ICG angiography with delayed graft function, describing a cut off ingress value of 106.23AU for the prediction of delayed graft function with a sensitivity of 78.3% and specificity of 80.8% (*p* < 0.0001) [9]. No further mention of the data related to the egress or egress rate was noted by Rother et al. [14].

### 3.11. Other

#### 3.11.1. Diabetic Wound Healing

Hajhosseini et al. researched the application of ICG angiography in the prediction of diabetic wound healing following hyperbaric oxygen therapy in cases and controls [21]. Absolute fluorescence intensity was selected as the perfusion parameter, showing a significant improvement between pre- and post-hyperbaric oxygen therapy in the patient group (*p* < 0.0015).

#### 3.11.2. Total Thyroidectomy

Parathyroid function and perfusion were intraoperatively measured by ICG angiography, assessing if the diagnosis could predict postoperative hypocalcemia [24]. Relative perfusion parameters based on the absolute fluorescent intensity and the average fluorescence intensity were selected, with the anterior trachea as the reference region. Relative absolute fluorescence intensity was predictive of both postoperative hypocalcemia (*p* = 0.027) and a postoperative drop in parathyroid hormone (*p* < 0.001). Relative average fluorescence intensity provided no significant results.

#### 3.11.3. Breast Imaging

Schneider et al. evaluated ICG imaging in the detection and characterization of breast lesions, namely malignant and benign [42]. The inflow parameters of peak amplitude (Imax) and time to peak (Tmax) were selected. A peak time-grouped amplitude was calculated, an average of the amplitudes in malignant and benign lesions at 30 time points. A significant difference in peak amplitude was noted between the two groups (*p* = 0.00015).

## 4. Discussion

This review highlights the broad selection of perfusion parameters in NIRF imaging. NIRF imaging shows great potential as a surgical tool, with applications in gastrointestinal, plastic, vascular, and neurological surgery. However, the quantification of NIRF imaging and the selection of optimal perfusion parameters faces several challenges.

The studies included in this review were largely performed in a small cohort setting, randomized clinical studies have yet to be performed. Research in this field is, to date, heterogeneous in study design, methodology, selected perfusion parameters, and endpoints. The heterogeneity in perfusion parameters selected and the large variation in the significance of outcomes in the articles presented in this review highlight the need for procedure-specific parameters and protocols to allow for the clinical application of this technique. Before the application of quantitative perfusion analysis in a clinical setting, comparison of the perfusion parameters in cases and controls is advised for baseline values.

While intensity dependent parameters are frequently discussed, they are susceptible to intra- and interpatient variability. Intensity dependent parameters can vary with changes in the imaging setting such as light intensity or camera angulation or distance from the patient. Statistical significance is often in a relative pre- and post-intervention setting, with no correlation to standard perfusion techniques such as the ankle–brachial index. Time-related parameters such as the Tmax or T1/2max are commonly mentioned as the parameters of choice, due to a reduced influence of fluorescence intensity as mentioned above. Furthermore, time-related parameters allow for the selection and measurement of regions of interest (ROIs) with a different camera distance and angle. This is advantageous in gastrointestinal surgery for example, where perfusion pre- and post-anastomosis needs to be documented.

Relative perfusion parameters are a favorable option for a clinical setting, to a degree, removing the influence of fluorescence intensity. Problems occur in a setting where the perfusion in the reference region is incorrectly assumed to be sufficient or is difficult to calculate. Peripheral arterial disease patients illustrate the pitfalls in relative parameters, with the example of a patient set to undergo an above the knee amputation. A large majority of patients also have arterial disease in the contralateral limb raising, making the reference perfusion value less reliable. Furthermore, should the patient already have a below the knee amputation of the intervention limb, it would not be feasible to film comparable sections of limb with one ICG dose. Table 3 summarizes the advantages and disadvantages of the various methods of quantification in NIRF imaging.

Limitations in this field concern the fluorescence angiography technique. In vascular surgery, the presence of inflammation or necrosis can impact the fluorescence signal. Inflammation leads to vasodilatation and hyperemia, falsely increasing the intensity-related parameters [43]. The influence of reactive hyperperfusion following a revascularization procedure in the field of vascular surgery also requires further research, as it remains unclear if post-revascularization NIRF measurements are representative of the patient’s vascular status. The influence of the circulatory status of the patient on NIRF parameters, such as Tmax or T1/2, is unclear. Theoretically, hypotension or reduced cardiac output could explain a lengthened Tmax, rather than the presence of an occlusion in neurovascular or vascular surgery. Furthermore, the impact of a patient’s plasma protein level on ICG uptake, and the subsequent intensity of the ICG measured requires further research.

The validity of quantified fluorescence data is further distorted by the range in ICG dose, camera selection, camera and environmental settings, and quantification software. Imaging systems have different sensitivities to fluorescence signals, creating heterogenous data before factoring in the impact of environmental influences. Son et al. selected an Image1 S fluorescence imaging system and (Tracker 4.97, Douglas Brown, Open Source Physics, Boston, MA, USA) quantification software, while Wada et al. chose a PDE-neo system and ROIs (Hamamatsu Photonics K.K.) analysis software [25,47]. The abovementioned differences between studies are explored in a recent systematic review by Lutken et al. [66]. The authors described the diversity in perfusion parameters and methodology in the field of gastrointestinal surgery, concluding that the application of ICG-NIRF imaging in this field requires standardization before implementation in a clinical setting. A potential means of achieving standardization is by the process of normalization, a mathematical means of correcting for fluctuations in fluorescence intensity, as yet only described in animal studies with promising results [67,68,69].

## 5. Conclusions

In conclusion, the quantitative analysis of fluorescence imaging has made great advances in recent years, showing potential across multiple surgical fields, both intraoperatively and in the relative pre- and postoperative evaluation of an intervention. Considering the rapidly growing application of this technique, research on the underlying quantification process is deficient. In this review, we have explored the perfusion parameters currently used in various surgical specializations. While this review highlights the heterogenicity in parameter selection, time-related parameters appear superior to absolute fluorescence intensity parameters in quantifying perfusion. Before ICG-NIRF imaging can be considered as the gold standard for perfusion quantification, the standardization of parameter selection and methodology with regard to ICG dose and camera type need to be explored.

## Figures and Tables

**Figure 1 life-11-00433-f001:**
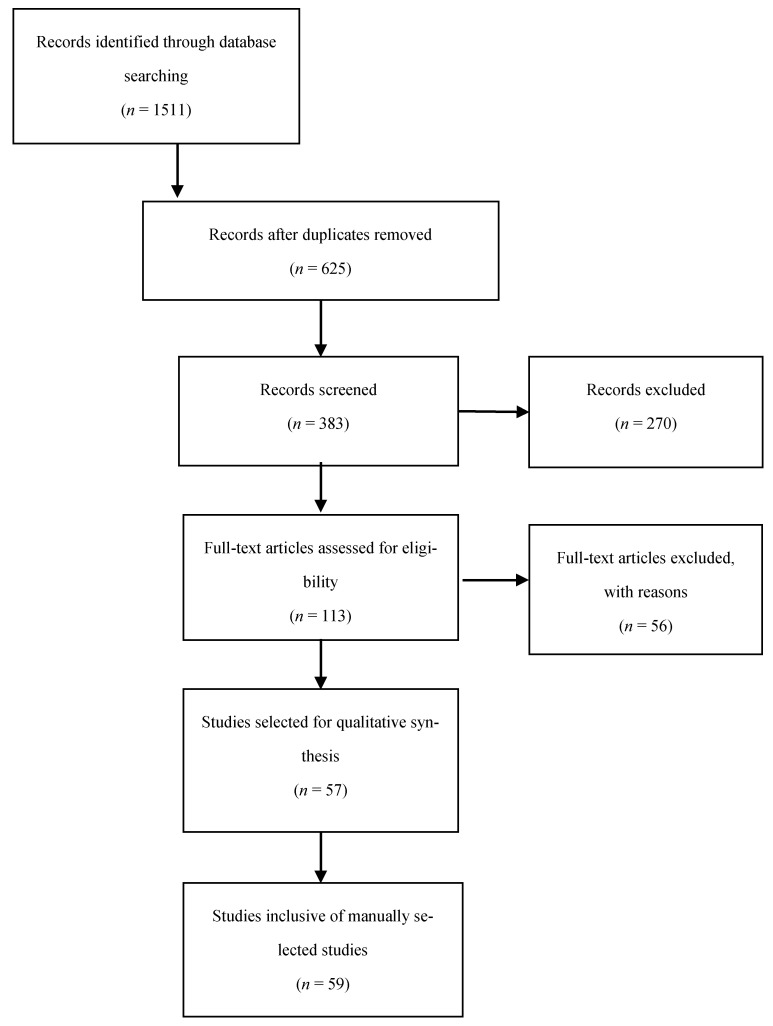
Preferred reporting items for systematic review and meta-analysis protocols flow chart for the selection of included studies.

**Figure 2 life-11-00433-f002:**
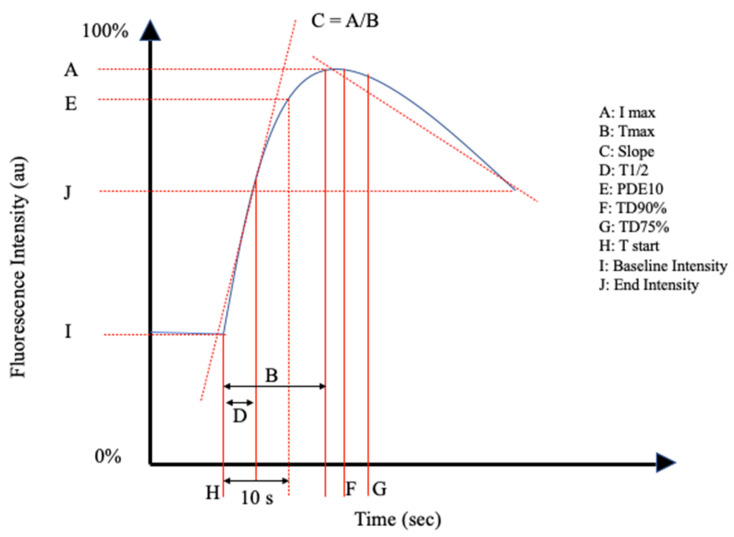
Schematic representation I of the perfusion parameters in Table 1.

**Figure 3 life-11-00433-f003:**
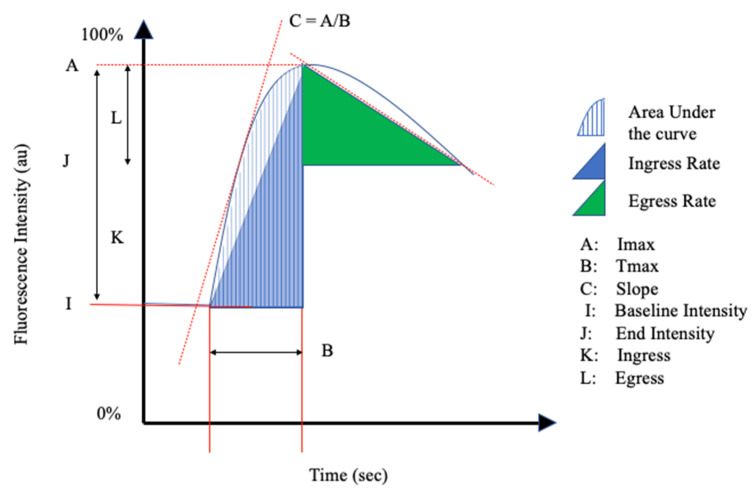
Schematic Representation II of the perfusion parameters in Table 1.

**Table 1 life-11-00433-t001:** Overview of fluorescence perfusion parameters.

Parameter	Definition	Equivalent	References
Ingress	Absolute difference between baseline fluorescence and its maximum value		[6,7,8,9,10,11,12,13,14,15,16,17]
Ingress Rate	Rate of increase of fluorescence signal from baseline to maximum value	Wash-in rate, fluorescence signal rise, blush rate	[8,9,10,11,12,13,14,15,16,17,18,19]
Ingress AUC	Area under the curve from baseline to maximum fluorescence intensity	WiAUC	[19]
Egress	Absolute difference between maximum intensity and the final intensity	Washout	[6,7,8,9,11,14,17,18,20]
Egress Rate	Rate of decrease of fluorescence signal from maximum value to the final intensity value		[8,9,11,14,17]
Fluorescence intensity	Fluorescence intensity		[8,21,22,23,24,25]
Imax	Maximum fluorescence intensity	Fmax, peak perfusion, FImax, MFI, cerebral blood volume	[6,7,24,25,26,27,28,29,30,31,32,33,34,35,36,37,38,39,40,41,42,43,44,45,46,47,48,49,50]
End Intensity	Fluorescence intensity at the end of the study	QEnd, residual FI	[8,33,51]
T start	Time to initial fluorescence signal	Tl (time local), latent time, T0, Te, TAP	[19,28,43,46,52,53,54,55]
Tmax	Time to maximum intensity	TTP (time to peak), blush time	[19,20,25,26,27,28,29,30,31,33,34,35,36,37,38,39,40,41,42,43,46,47,49,53,56,57]
Delta T	Time from initial fluorescent intensity to Imax	Tmax	[26,55]
T 1/2	Time to half of the maximum fluorescence intensity	Tmax1/2, delay	[25,27,28,29,30,32,33,34,36,38,39,41,47,48,58,59,60]
TR	Time ratio (T1/2/Tmax)		[25]
Rise Time	Time from (10–90%) OR (20–80%) of maximum fluorescence intensity OR time from 20–80% of maximum fluorescence intensity		[19,33,34,35,41,45,48,61]
Td90%	Time from Fmax to 90% of the Fmax		[30,39]
Td75%	Time from the Fmax to 75% of the Fmax		[30]
IR 60 s	The rate of intensity measured 60 s after the Tmax to the Fmax (intensity at 60 s after Tmax/Fmax)		[30]
Wash-in perfusion index	Ratio between the WiAUC to the rise time		[19]
Slope	Fmax/Tmax	BFI (blood flow index), perfusion rate, Smax, cerebral blood flow, perfusion Index	[25,28,29,37,40,41,43,44,46,47,48,49,50,51,52,56,57]
BFI	Blood flow index (Fmax/RT) OR (F90-F10)/(T90-T10)	Slope	[23,33,34,35,41,45,48,58,59,62]
S 1/2	Slope of the intensity increase from baseline to half the maximum intensity		[37]
PDE10	The fluorescence intensity increase at 10 s	SPY10	[29,37,44,60]
ITT	Intrinsic transit time—the time needed for the fluorescent dye to circulate from arterial to venous anastomosis	Transit time	[23,27,41,45,48,61]
AUC	Area under the curve of intensity over time	Curve integral	[8]
Perfusion rate	Fraction of blood exchanged per min in vascular volume (%/min)		[20]
Relative perfusion	Perfusion as a percentage of a reference region		[6,7,24,50,51,52,53,57]

**Table 2 life-11-00433-t002:** Overview of study characteristics and results of near infra-red fluorescence imaging.

Application	Reference	Patients	Study Characteristics		
			Camera	Software	ICG Dose
Gastro-intestinal	Aiba [54]	110	OPAL1	Not spec	0.1 mg/kg
Surgery	Amagai [26]	69	Karl Storz	ImageJ	0.2 mg/kg
	D’Urso [56]	22	D-Light P	FLER	0.2 mg/kg
	Hayami [28]	22	D-Light P	Hamamatsu Photonics	5 mg/2 mL
	Ishige [31]	20	Olympus	Hamamatsu Photonics	1.25 mg
	Kamiya [32]	26	PDE Hamamatsu	Hamamatsu Photonics	1 mL
	Son [25]	86	Image1	Tracker 4.97	0.25 mg/kg
	Wada [47]	112	PDE Hamamatsu	Hamamatsu Photonics	5 mg
Neurosurgery	Goertz [27]	54	Carl Zeiss Co.	Flow 800	10 mg
	Holling [61]	5	OPMI Pentero Microscope	Flow 800	5 mg
	Kamp [33]	10	OPMI Pentero Microscope	Flow 800	5 mg
	Kamp [34]	30	OPMI Pentero Microscope	Flow 800	5 mg
	Kobayashi [35]	10	OPMI Pentero Microscope	Flow 800	7.5 mg/3 mL
	Prinz [58]	30	OPMI Pentero Microscope	Flow 800	0.25 mg/kg
	Rennert [41]	7	OPMI Pentero Microscope	Flow 800	0.2 mg/kg
	Rennert [59]	10	OPMI Pentero Microscope Or Kinevo	Flow 800	0.2 mg/kg
	Shi [45]	9	OPMI Pentero Microscope	Flow 800	0.1 mg/kg
	Uchino [63]	10	OPMI Pentero Microscope	Flow 800	0.1 mg/kg
	Uchino [64]	7	OPMI Pentero Microscope	Flow 800	0.1 mg/kg
	Woitzik [62]	6	IC-View	IC Calc	0.3 mg/kg
	Ye [48]	87	Carl Zeiss Co.	Flow 800	0.25 mg/kg
	Zhang [49]	60	Not spec	Flow 800	Not spec
Plastic Surgery	Abdelwahab [6]	71	SPY Elite Imaging System	SPY-Q	5 mg/2 mL
	Abdelwahab [7]	10	SPY Elite Imaging System	SPY-Q	5 mg/2 mL
	Betz [52]	11	Karl Storz	IC Calc	0.3 mg/kg
	Betz [53]	25	ICG Pulsion	IC Calc	0.3 mg/kg
	Fichter [19]	40	Pulsion PDE	ImageJ	0.3 mg/kg
	Girard [10]	40	SPY Elite Imaging System	SPY-Q	5 mg
	Gorai [57]	181	PDE Hamamatsu	Hamamatsu Photonics	25 mg/2 mL
	Han [22]	32	SPY Elite Imaging System	Not spec	2.5 mg
	Hitier [23]	20	Fluobeam	Fluobeam	0.25 mg/kg
	Maxwell [36]	1	SPY Elite Imaging System	Not spec	Not spec
	Miyazaki [37]	8	PDE Hamamatsu	Hamamatsu Photonics	0.1 mg/kg
	Mothes [51]	35	IC-View	IC Calc	0.5 mg/kg
	Rother [13]	23	SPY Elite Imaging System	SPY-Q	0.1 mg/kg
	Tanaka [46]	8	PDE Hamamatsu	Hamamatsu Photonics	0.1 mg/kg
	Yang [17]	10	SPY Elite Imaging System	SPY-Q	3 mL
Vascular	Braun [8]	24	SPY Elite Imaging System	Not spec	Not spec
	Colvard [18]	93	SPY Elite Imaging System	SPY-Q	2.5 mL
	Igari [29]	21	PDE Hamamtsu	Hamamatsu Photonics	0.1 mg/kg
	Igari [30]	23	PDE Hamamtsu	Hamamatsu Photonics	0.1 mg/kg
	Kang [55]	2	Vieworks	Visual C++	0.16 mg/kg
	Kang [20]	2	VIeworks	Not spec	0.16 mg/kg
	Mironov [11]	28	SPY Elite Imaging System	SPY-Q	5 mg/250 mL
	Nakamura [38]	21	PDE Hamamatsu	Hamamatsu Photonics	0.1 mg/kg
	Nishizawa [39]	62	PDE Hamamatsu	Hamamatsu Photonics	0.1 mg/kg
	Patel [40]	47	SPY Elite Imaging System	Not spec	0.1 mg/kg
	Regus [12]	47	SPY Elite Imaging System	SPY-Q	0.002 mg/kg
	Rother [15]	40	SPY Elite Imaging System	SPY-Q	0.1 mg/kg
	Rother [16]	33	SPY Elite Imaging System	SPY-Q	0.1 mg/kg
	Seinturier [43]	34	Fluobeam	Not spec	0.05 mg/kg
	Settembre [44]	101	SPY Elite Imaging System	Not spec	0.1 mg/kg
	Terasaki [60]	34	PDE Hamamatsu	Hamamatsu Photonics	0.1 mg/kg
	Venermo [65]	41	PDE Hamamatsu	Hamamatsu Photonics	0.1 mg/kg
	Zimmermann [50]	30	IC-View	IC-Calc	0.5 mg/kg
Transplantation	Rother [14]	77	SPY Elite Imaging System	SPY-Q	0.02 mg/kg
	Gerken [9]	128	SPY Elite Imaging System	SPY-Q	0.02 mg/kg
Thyroid Surgery	Lang [24]	70	SPY Elite Imaging System	Not spec	2.5 mg
Diabetic Foot	Hajhosseini [21]	21	LUNA Fluorescence Microscope	SAS	5 mg/mL
Breast lesions	Schneider [42]	30	NIRx Medical Technologies	NIRx NAVI	2.5 mg

**Table 3 life-11-00433-t003:** Summary of methods of quantification in NIRF imaging.

	Advantages	Disadvantages
Intensity-related parameters	Broad parameter selection	Influencing factors on intensity: - Patient-related: ICG concentration, cardiac output - System-related: camera distance, camera angle, environmental light
Time-related parameters	No influence of measured intensity Comparison possible between ROIs with different camera distance and angle	Narrow parameter selection
Relative parameters	Patient vasculature provides case–control data	Reference region may not be representative of optimal perfusion Influencing factors on intensity (see above)

## Data Availability

No new data were created or analyzed in this study. Data sharing is not applicable to this article.

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
