# Peer review of "Perfusion Parameters in Near-Infrared Fluorescence Imaging with Indocyanine Green: A Systematic Review of the Literature"

_life, 2021, doi:10.3390/life11050433_

Round 1

Reviewer 1 Report

I read with interest Lauren N. Goncalves’ article concerning a systematic review of NIRF perfusion parameters with Indocyanine green.

The authors perform a systematic review on the use of NIRF imaging technique to asses tissue perfusion in various surgical fields focusing on the widely heterogeneous parameters correlating fluorescence images to tissue perfusion.

This subject is interesting, extremely topical and widely explored in many different surgical specialties.

As largely mentioned in the article actually literature lacks of a univocal largely accepted parameter to best assess tissue perfusion. 

Comments:

  1. The conclusion of the article is that “time-related parameters” appear superior to absolute intensity but this seems not to be clearly demonstrated and seems just an opinion of the authors. Up to me why time-related parameters seems to be superior should be discussed in more details into the discussion section and supported with some clear evidence.
  2. In order to improve this first point authors could add a table resuming and comparing the advantages of these two different parameters “time-related parameters” VS “absolute intensity”.  
  3. In the introduction section authors mentioned the various surgical fields that have explored NIRF imaging to asses tissue perfusion: plastic, vascular, cardiac, neuro, gastrointestinal surgeries but doesn’t mention at all endocrine surgery (is mentioned only once in results section - line370). This field should also be explored in the introduction since there are a lot of articles studying the perfusion of parathyroid glands after thyroidectomy (with promising effects on the reduction of post-operative hypoparathyroidism) and on remnant perfusion in partial adrenal gland resection.
  4. Line 86: you have included also studies with 1 patient included but probably these studies should be excluded since not really significant(s).
  5. Table 1. Seems not essential in the main document and should be provided in supplement material
  6. Figure 1. Exclusion criteria should be mentioned. Why did you exclude 270 and 56 articles this information seems not been given in the text.
  7. Figure 3 and 4 are not mentioned in the text and should also be explained in the discussion section.
  8. Line 216 doubled “in”

Reviewer 2 Report

Very interesting topic, the good organization of the manuscript, its sonority, and its easy reading, make it captivatinf for the reader. Adequate and recent references, well illustrated tables, good research method. Manuscript can be accepted for pubblication

Author Response

We thank you for your valuable time and useful contribution regarding our systematic review.

Reviewer 3 Report

The authors conducted a very interesting systematic review on perfusion parameters during near infrared fluorescence imaging with indocyanine green. From my point of view they address an important topic in this field of imaging, as there is currently a lack of comparability and standardization in the use of these perfusion parameters. Even in the different fields of medicine such as vascular surgery or general surgery, the different results of the studies are not comparable, as everybody uses different quantification parameters.

I have a few comments.

Introduction:

  • Very sound, maybe transplant surgery should also be mentioned in the introduction, as you separate later accordingly.

Methods

  • Search strategy: did you use grey literature?
  • You adhered to the PRISMA guidelines as you mentioned in the results section, this should be included and cited in the methods section accordingly
  • Was your study registered (a.e. Prospero?)

Results

  • Figure 1: n= is missing for the initial found articles

Discussion:

  • Maybe some more current limitations of ICG fluorescence angiography with regards to quantification should be mentioned:
    • Hyperperfusion (a.e. vascular surgery) due to inflammation/infection
    • Unknown influence of plasma protein level on ICG uptake
    • Influence of circulatory situation during operation, for example use of catecholamine
    • Influence of reactive hyperperfusion due to revascularization

Round 2

Reviewer 1 Report

Thank you for the revision. 

No more comments.